# BRAF/EZH2 Signaling Represses miR-129-5p Inhibition of SOX4 Thereby Modulating BRAFi Resistance in Melanoma

**DOI:** 10.3390/cancers13102393

**Published:** 2021-05-15

**Authors:** Kathleen Gebhardt, Bayram Edemir, Elisabeth Groß, Linda Nemetschke, Stefanie Kewitz-Hempel, Rose K. C. Moritz, Cord Sunderkötter, Dennis Gerloff

**Affiliations:** 1Department of Dermatology and Venereology, Martin-Luther-University Halle-Wittenberg, 06120 Halle (Saale), Germany; kathleen.gebhardt@uk-halle.de (K.G.); linda.nemetschke@uk-halle.de (L.N.); stefanie.kewitz-hempel@uk-halle.de (S.K.-H.); rose.moritz@uk-halle.de (R.K.C.M.); cord.sunderkoetter@uk-halle.de (C.S.); 2Department of Internal Medicine IV, Hematology and Oncology, Martin-Luther-University Halle-Wittenberg, 06120 Halle (Saale), Germany; bayram.edemir@uk-halle.de (B.E.); elisabeth.gross@uk-halle.de (E.G.)

**Keywords:** melanoma, BRAF mutation, miRNAs, therapy resistance

## Abstract

**Simple Summary:**

Approximately 60% of all melanomas are associated with a constitutive activating BRAF mutation. Inhibition of BRAF downstream signaling by targeted therapies significantly improved patient outcomes. However, most patients eventually develop resistance. Here we identified miR-129-5p as a novel tumor suppressor in BRAF mutated melanoma, which expression is increased during response to BRAF inhibition, but repressed in an EZH2 dependent manner during activated BRAF signaling. Overexpression of miR-129-5p decreases melanoma cell proliferation and improves response to BRAF inhibition by targeting *SOX4*. Taken together our results emphasize SOX4 as a potential therapeutic target in BRAF driven melanoma which could be attacked by pharmaceutically.

**Abstract:**

Many melanomas are associated with activating BRAF mutation. Targeted therapies by inhibitors of BRAF and MEK (BRAFi, MEKi) show marked antitumor response, but become limited by drug resistance. The mechanisms for this are not fully revealed, but include miRNA. Wishing to improve efficacy of BRAFi and knowing that certain miRNAs are linked to resistance to BRAFi, we wanted to focus on miRNAs exclusively associated with response to BRAFi. We found increased expression of miR-129-5p during BRAFi treatment of BRAF- mutant melanoma cells. Parallel to emergence of resistance we observed mir-129-5p expression to become suppressed by BRAF/EZH2 signaling. In functional analyses we revealed that miR-129-5p acts as a tumor suppressor as its overexpression decreased cell proliferation, improved treatment response and reduced viability of BRAFi resistant melanoma cells. By protein expression analyses and luciferase reporter assays we confirmed *SOX4* as a direct target of mir-129-5p. Thus, modulation of the miR-129-5p-SOX4 axis could serve as a promising novel strategy to improve response to BRAFi in melanoma.

## 1. Introduction

Melanoma is the most lethal form of skin cancer with an increasing incidence [1]. Stage IV melanoma has a poor prognosis for patients, with a 5-year survival probability of less than 5–25%, if untreated [2,3]. Progression of a majority of cutaneous melanomas depends on oncogenic, partially mutually exclusive somatic mutations involving BRAF (50–60%), NRAS (20–26%), TP53 (19%), or PTEN (12%) [4,5,6,7,8]. The most common BRAF mutation is the V600E (90%) substitution, which constitutively activates aberrant BRAF signaling [9].

During the last decade, BRAF-mutant melanomas have become amenable to new targeted therapies based on initially BRAF inhibitor (BRAFi) and now on its combination with a MEK inhibitor (MEKi). Introduction of combinatory BRAFi/MEKi therapies improved patient outcomes significantly [10,11,12]; however, most patients eventually develop resistance mechanisms towards these targeted therapies [13]. Several molecular mechanisms are involved in the acquisition of BRAFi resistance. The most frequent one encompasses reactivation of the MAPK pathway, commonly through NRAS mutations [14], alterations in BRAF splicing [15] or amplification [16,17] and less often by alterations of MEK1/2 [18]. Alternatively, in some patients, the PI3K/AKT pathway, a secondary signaling route becomes hyper-activated [19]. Adaptive resistance to MEKi likewise is acquired by reactivation of MAPK signaling (e.g., ERK) or activation of parallel signaling pathways (e.g., PI3K, STAT and Hippo signaling pathways) [20]. Yet, a considerable proportion of BRAFi resistant tumors (40%) displays mechanisms of resistance that have not been fully revealed [21,22].

To identify novel strategies for improving efficacy of BRAFi in melanoma, it would be helpful to find molecular mechanisms of resistance, which can be modified or reverted.

There is evidence that miRNAs are involved in the development of resistance in melanoma treatment to targeted therapies [23]. MicroRNAs are small noncoding RNAs (20–22 nt) which post-transcriptionally regulate protein expression. To this end miRNAs guide the RNA-induced silencing complex (RISC) to a complementary seed sequence in the 3′untranslated region (3′UTR) of target mRNAs [24]. Binding of the miRNA-RISC reduces the efficacy of protein translation and induces destabilization and cleavage of the targeted mRNA [25]. MicroRNAs are involved in development or progression of cancer, but obviously also in mechanisms of drug resistance in leukaemia and in various solid cancers [26,27,28,29,30]. They could present potential therapeutic targets, by the use of chemically modified complementary RNA molecules, such as locked nucleic acids (LNAs) or miRNA mimics [31,32,33,34].

Thus, there is promise to investigate this group of molecular factors in more detail to define novel mechanisms of resistance, which could later on serve as potential therapeutic targets.

In melanoma, the expression of miR-7 [35], miR-126-3p [36] and miR-579-3p [37] is significantly reduced in BRAFi resistant cell lines, while their overexpression increases the sensitivity to BRAFi. In contrast, miR-34a, miR-100 and miR-125b were highly expressed in resistant melanoma cells and inhibition of those miRNAs restores the sensitivity of BRAFi resistant cells [38]. The miRNA family miR-204-5p and miR-211-5p are the most investigated, but most controversially discussed miRNAs associated with tumor progression and BRAFi resistance in melanoma. Both are induced by BRAFi treatment and their expression remains increased in cells resistant to BRAFi [39,40,41]. Moreover, enforced expression miR-211-5p contributes to BRAFi resistance by directly targeting DUSP6, which modulates the DUSP6-ERK5 signaling axis and promotes BRAFV600E driven melanoma growth and BRAFi/MEKi inhibitor resistance [40]. Vitiello et al. reported a context dependent function of miR-204 and miR-211. They show that in melanotic melanoma miR-211 targets EDEM1 [41], which potentiates Vemurafenib induced pigmentation, resulting in a limited Vemurafenib efficacy. miR-204 was shown to inhibit melanoma cell motility by targeting AP1S in amelanotic melanoma [41].

These findings illustrate that specific miRNAs are involved in alternative mechanisms of BRAFi resistance and that their manipulation can partially restore sensitivity to BRAF targeted therapies.

While microRNAs have been linked mostly to resistance to BRAFi in melanoma, we wanted to identify miRNAs associated specifically with response to BRAFi. We found that miR-129-5p was most strongly induced after Vemurafenib treatment in parental BRAF mutated melanoma cells, but not in the corresponding BRAFi resistant cells, normal melanocytes or BRAF wildtype melanoma cells. Subsequently, we investigated the transcriptional regulation of miR-129-5p downstream of constitutive active BRAF signaling and the biological function of miR-129-5p in BRAF mutated melanoma, as well as BRAFi resistance. Further, we wanted to identify a direct target of miR-129-5p, which mediates melanoma progression and BRAFi resistance.

The results of our study demonstrate that EZH2 dependent repression of miR-129-5p is solved by BRAFi/MEKi treatment thereby modulating BRAFi resistance and melanoma progression via targeting SOX4.

## 2. Materials and Methods

### 2.1. Data Sets

The published datasets GSE94423 [41], GSE98314 and GSE50509 [22] were obtained from GEO DataSets (https://www.ncbi.nlm.nih.gov/gds accessed on 22 April 2021). For GSE98314 and GSE50509 gene expression data were normalized using the cubic spline function. TCGA expression data (log2(RPM + 1) of miR-129-5p and EZH2 mutation status of melanoma samples were used from the cancer genome atlas (https://www.cancer.gov/tcga accessed on 22 April 2021).

### 2.2. Cell Culture

Normal human epidermal melanocytes (NHEM) were isolated from foreskin tissues and cultured in medium 254 (Cascade Biologics^®^) including human melanocyte growth supplement (HMGS) and 1% penicillin-streptomycin. Melanoma cell lines (A375, WM35, WM902B, WM9, MV3 and SK-Mel30) were cultured in DMEM supplemented with 10% fetal calve serum (FCS) and 1% penicillin-streptomycin. All cells were incubated at 37 °C and 5% CO_2_. Resistant cell lines A375R, WM35R, WM902BR and WM9R were generated by treating parental BRAFV600E mutant melanoma cell lines with increasing concentrations of Vemurafenib (PLX4032; LC-Laboratories, Woburn, MA, USA). Cells with the ability to grow in 2 µM or in 10 µM Vemurafenib were described as resistant and maintained in continuous presence of 2 µM or 10 µM Vemurafenib. For MEK inhibitor treatment we used Trametinib (LC-Laboratories, Woburn, MA, USA) with indicated concentrations. EZH2 inhibition was performed by Tazemetostat (EPZ-6438) (Selleckchem, Houston, TX, USA).

### 2.3. miRNA Detection by Quantitative Real-Time PCR

Total RNA was extracted from cells using the TriFast™ reagent (Peqlab). The miRNA quantification was performed by qRT-PCR using TaqMan^®^ MicroRNA Reverse Transcription Kit and TaqMan^®^ Universal Master Mix II following the manufacturer’s instructions. For normalization RNUB6 expression was used. TaqMan^®^ miRNA assays for RNUB6 and hsa-miR-129-5p were obtained from Thermo Fisher Scientific.

### 2.4. Transcriptional Analysis by qRT-PCR

The total RNA was isolated using TriFast™ reagent (Peqlab). Reverse transcription was performed by M-MLV Reverse Transcriptase using oligo(dT)_18_-primer (Thermo Fisher Scientific, Waltham, MA, USA). For quantitative RT-PCR we used PowerUp™ SYBR™Green master mix (Thermo Fisher Scientific, Waltham, MA, USA) following the manufacturer’s instructions. qPCRs were performed on QuantStudio™ 5 Real-Time PCR Systems. Real-time quantitative PCR assay was performed to detect the relative expression level of SOX4 (forward 5′-GGTCTCTAGTTCTTGCACGCTC-3′ and reverse 5′-CGGAATCGGCACTAAGGAG-3′) with GAPDH (forward 5′-ACCACAGTCCATGCCATCAC-3′ and reverse 5′-TCCACCACCCTGTTGCTGTA-3′) as endogenous control. The relative expression levels were calculated with the 2^−∆∆Ct^ method and experiments were repeated at least in three independent triplicates.

### 2.5. DNA Constructs and siRNA

For miR-129-5p overexpression miR-129 sequence was cloned into the pcDNA6.2-GW/EmGFP-miR plasmid using the BLOCK-iT Pol II miR RNAi Expression Vector Kit (Invitrogen). The following oligonucleotide sequences were used: miR-129-top, 5′-TGCTGCTTTTTGCGGTCTGGGCTTGCGTTTTGGCCAC TGACTGACGCAAGCCCAGAGCAAAAAG-3′ and miR-129-bottom, 5′-CCTGCTTTTTGCTCTGGGCTT GCGTCAGTCAGTGGCCAAAACGCAAGCCCAGACCGCAAAAAG-3′. The correct assembly of the vectors was verified by sequencing. pmiRZip-129-5p construct (Cat# MZIP129-5p-PA-1) for miR-129-5p inhibition and control vector pmiRZip-scr (Cat#MZIP000-PA-1) were obtained from System Biosciences (Mountain View, CA, USA). SOX4 siRNA was obtained from Qiagen (Hilden, Germany).

### 2.6. Transfections

Transfection to A375 and WM35 cells was performed with Lipofectamine 3000 reagent (Invitrogen) or Amaxa Cell line Nucleofector Kit V (LONZA) following the manufacturer’s instructions. Positive transfected cells were sorted by FACS, blasticidin (pCDNA6.2-constructs) or puromycin (miRZIP-constructs) selection.

### 2.7. Immunoblot Snalyses

Cells were lysed using a RIPA buffer for 30 min at 4 °C. Protein extracts were resolved by SDS–PAGE, blotted to nitrocellulose membranes and probed with the following antibodies: anti-GAPDH (Cat#2118), anti-EZH2 (Cat#5246), anti-H3 (Cat#4499), anti-H3K27me3 (Cat#9733), anti-ERK1/2 (Cat#4695) and anti-phosphoERK1/2 (Cat#4377) all from Cell Signaling Technology (Danvers, MA, USA) and anti-SOX4 (Cat#LS-C499849) from LSBio (Eching, Germany). For antibody detection we used anti-rabbit IgG-HRP (Cat#7074, Cell Signaling Technology, Danvers, MA, USA).

### 2.8. Cell Viability Assay

The number of viable cells was determined by CellTiter-Glo^®^ Luminescent Cell Viability Assay (Promega). Cells (5000 per well) were seeded in 96-weell plates. After 24 h the cells were treated with 0.4 µM Vemurafenib or DMSO as control for at least 72 h. Cell viability was documented by TECAN plate reader following the manufacturer’s instructions.

### 2.9. Cell Growth Assay

For growth curves, 1 × 10^5^ of respective cells were seeded in each well of a 6-well plate and cell numbers were determined over the time periods indicated by using a Neubauer counting chamber.

### 2.10. Cell Cycle Analysis

For cell cycle analysis, cells were trypsinized and fixed with 70% ice-cold ethanol for 30 min on 4 °C, washed twice with PBS and resuspended in 200 µL PI (from 50 µg/mL stock solution, abcam). Stained cells were analyzed on a BD FACS Scan cytometer using CellQuest software (BD Biosciences, Franklin Lakes, NJ, USA).

### 2.11. 3D Spheroid Growth Assay

For melanosphere formation, 5000 cells were seeded in a 3D culture-qualified 96-well spheroid formation plate (Cat#650970, Greiner). Spheroids were formed for 96 h before treatment. The growth of 3D spheroid cultures was assessed 96 h after seeding (0 h) and a following initial Vemurafenib treatment (1 µM) for 72 h, 96 h and 144 h. Melanospheres were photographed at indicated time points with a Keyence BZ-X810 fluorescence microscope and sphere areas were determined using ImageJ software.

### 2.12. Luciferase Reporter Assay

To confirm miR-129-5p binding to SOX4 3′UTR we used luciferase reporter clone for human SOX4 3′UTR (HmiT017630-MT06, Genecopoeia) and luciferase control reporter construct (CmiT000001-MT06, Genecopoeia). Constructs were transfected in A375 (ctrl) and A375 miR129-5p (overexpression) cells 24 h after seeding. The relative Luciferase activity was measured 48 h after transfection using Luc-Pair Duo-Luciferase Assay Kits 2.0 (Genecopoeia) following the manufacturer’s instructions.

### 2.13. Statistical Analyses

Statistical analyses were performed with GraphPad Prism. Student’s t-test was used for independent or paired samples to determine the statistical significance of experimental results. A *p*-value of 0.05 or less was considered significant. The results were represented as the average ± standard deviation from at least three independent experiments.

## 3. Results

### 3.1. Expression of miR-129-5p Increases during BRAFi and MEKi Treatment

To identify miRNAs associated with response or resistance to BRAFi and MEKi treatment we reanalyzed a published next generation data set (GSE94423) [41]. To this end, we compared the miRNA expression of parental A375 cells treated with Vemurafenib to three different conditions: parental A375 cells treated with DMSO, a resistant clone of A375 cells (A375R) treated with Vemurafenib and A375R treated with DMSO. In our analysis we found 71 miRNAs upregulated (log2 fold change >1; *p* ≤ 0.05) and 137 miRNAs downregulated (log2 fold change <−1; *p* ≤ 0.05) in parental A375 cells treated with Vemurafenib (Figure 1A). One of the miRNAs which were induced the most after Vemurafenib treatment was miR-129-5p (Figure 1A). We could confirm our data analysis by virtue of a second published dataset (GSE98314) encompassing 11 BRAF mutated cell lines treated with BRAFi (Dabrafenib) or BRAFi/MEKi (Dabrafenib/Trametinib) and DMSO treated controls: when we performed a comparative analysis, we found that miR-129 expression is induced under BRAFi or BRAFi/MEKi in 10 out of 11 cell lines (Figure 1B).

In order to confirm these in silico findings we performed qRT-PCR of BRAF mutated, treatment sensitive melanoma cell lines (A375, WM9, WM35 and WM902B) and found that miR-129-5p expression was increased after Vemurafenib treatment. The respective BRAF resistant cell lines (A375R, WM9R, WM35R and WM902BR (Appendix A), as well as BRAF wildtype melanoma cell lines (MV3, SK-Mel30) and normal human epidermal melanocytes (NHEM) showed no significant change in miR-129-5p expression after Vemurafenib treatment (Figure 1C).

Further analyses demonstrated that expression of miR-129-5p was additionally increased during treatment with MEKi (Trametinib) (Figure 1D) also in BRAFi resistant cell lines (A375R, WM35R) (Figure 1D). Combinatory treatment with Vemurafenib and Trametinib also induces the expression of miR-129-5p (Appendix A).

Taken together these results strongly suggest that miR-129-5p expression is mediated by BRAF/MEK pathway signaling.

### 3.2. miR-129-5p Expression Decreases During Emergence of Resistance to BRAFi

Having demonstrated that miR-129-5p expression is mediated by constitutive active BRAF/MEK signaling in BRAF mutated melanoma, we investigated its expression during long term BRAFi or MEKi treatment. Therefore, A375 and WM35 cell lines were treated with Vemurafenib or Trametinib for a periode of 20 days. Culture medium containing the inhibitors was exchanged every 48 h.

The treatment initially resulted in morphological changes of the melanoma cells: Initially both BRAFi and MEKi treated cells became spindle-shaped and ceased to proliferate, reflecting the antineplastic response. Then cells treated with BRAFi regained initial morphology and increased proliferation at days 10–15, indicating emerging resistance to treatment. Trametinib treated cells showed a prolonged treatment response (Figure 2A, Appendix A). Next, we analyzed the expression of miR-129-5p at specific time points by qRT-PCR. Expression of miR-129-5p strongly increased in the first days during treatment with Vemurafenib and declined to almost the initial expression levels at day 20 (Figure 2B); this time course correlated with the morphological changes and the initially interrupted and finally resumed proliferation of the cells reflecting emergence of resistance.

Trametinib treatment delivered similar results, only that induction of miR-129-5p expression was stronger compared to BRAFi treatment and that the decrease of miR-129-5p expression did not reach the initial level by day 20 of treatment (Figure 2C). When we investigated Vemurafenib resistant cell lines (A375R, WM35R), we observed a reduced expression of miR-129-5p compared to parental cell lines (A375, WM35) (Figure 2D). We could evaluate this association of miR-129-5p levels with the response to Dabrafenib or Vemurafenib also found in the published data set (GSE50509) of melanoma tumor samples from patients before starting Dabrafenib or Vemurafenib and at the time of tumor progression: here, the levels of miR-129-5p in untreated melanoma samples were equal to samples of treatment resistant and progressive melanoma (Figure 2E). Our results demonstrate that miR-129-5p expression is induced in cells responsive to Vemurafenib or Trametinib treatment, but inhibited during treatment resistance.

### 3.3. EZH2 Suppresses miR-129-5p Expression Downstream of Constitutive Active BRAF Signaling

Since miR-129-5p expression in various cancer entities (e.g., endometrial cancer, breast cancer and gastric cancer) is mediated by EZH2 [42,43,44], an epigenetic modulator of H3K27me3 and DNA methylation [45,46], we investigated if EZH2 is mediated by BRAF/MEK pathway inhibition. Western blot analyses revealed that BRAFi (Vemurafenib), as well as MEKi (Trametinib), treatment decreased EZH2 protein levels in BRAF mutated cell lines (A375, WM35) (Figure 3A). We also demonstrated that even in BRAFi resistant cell lines (A375R, WM35R), MEK inhibition reduced EZH2 protein expression, while BRAF inhibition did not (Figure 3A). Consequently, we analyzed if inhibition of EZH2 induces miR-129-5p expression in BRAF mutated cell lines. Inhibition of EZH2 with the specific inhibitor EPZ-6438 (EPZ) significantly increased miR-129-5p expression in BRAF mutated melanoma cell lines (A375, WM35), even in BRAFi resistant cell lines (A375R, WM35R). In comparison, normal melanocytes (NHEM) and BRAF wildtype melanoma cell lines (MV3, SK-Mel30) displayed no induction of miR-129-5p expression (Figure 3B). Reanalysis of a published dataset (GSE98314) revealed that treatment with the BRAFi Dabrafenib decreased EZH2 expression in 11 different BRAF mutated melanoma cell lines (Figure 3C) and we found a significant inverse correlation of EZH2 and miR-129 expression (r = −0.46; *p* = 0.029) (Figure 3D). Additionally, the subset of the BRAF mutated melanoma cohort of the cancer genome atlas (TCGA) harboring an EZH2 silent mutation, showed a highly significant enforced miR-129-5p expression compared to BRAF mutated melanoma patients with EZH2 wildtype (Figure 3E).

To confirm that EZH2 is involved in transcriptional regulation of miR-129-5p, we analyzed H3K27me3 of Vemurafenib treated A375 and WM35 cells. Because we observed no changes in H3K27me3 after BRAFi treatment (Appendix A), we investigated if miR-129-5p expression is controlled by DNA methylation, an epigenetic mechanism mediated by EZH2. Therefore, we analyzed miR-129-5p expression after DNA methylation inhibition by decetabine (DAC) and found a significant miR-129-5p induction in BRAF mutated A375 cells (Figure 3F).

Taken together these results indicate that EZH2 mediates miR-129-5p expression downstream of constitutive active BRAF signaling.

### 3.4. miR-129-5p Acts as Tumor Suppressor In Vitro and in a 3D Spheroid Model

Having established that miR-129-5p is induced during BRAFi and MEKi response and mediated by EZH2 downstream of constitutive active BRAF signaling, we subsequently studied the biological function of miR-129-5p on proliferation of BRAF mutated melanoma cells: miR-129-5p overexpression reduced proliferation of melanoma cell line A375, whereas knockdown of miR-129-5p by a miRZip-129-5p construct significantly increased proliferation (Figure 4A). Since, miR-129-5p expression was upregulated during BRAFi or MEKi treatment in sensitive, but not in resistant cells, we wondered if regulation of miR-129-5p could mediate potentially therapeutic effects in terms of overcoming resistance. Therefore, we performed viability assays for A375 cells with either knockdown or overexpression of miR-129-5p during BRAFi treatment. MiR-129-5p knockdown increased viability of Vemurafenib treated A375 cells (A375 scr IC_50_: 0.25 µM vs. A375 miRZip-129 IC_50_: 1.15 µM) (Figure 4B), while its overexpression improved the treatment response of BRAFi resistant A375R cells (A375R scr IC_50_: 4.37 µM vs. A375R miR-129-5p IC_50_: 3.87 µM) (Figure 4C). In cell cycle analyses we found, that inhibition of miR-129-5p function through expression of a miRZip-129 construct in A375 cells increased the number of cells in the S/G2 phase during Vemurafenib treatment compared to A375 scr (Figure 4D). Interestingly, the cell cycle distribution of A375 miRZip-129 under BRAFi treatment cells was similar to A375R cells (Figure 4D). In contrast, untreated A375 miRZip-129 and scr cells showed an equivalent cell cycle that was different from A375R cells (Appendix A).

To approach the question if such effects are relevant in situ we used a 3D spheroid model. While A375 scr control cells and miR-129-5p knockdown cells formed intermediate ragged spheroids, cells overexpressing miR-129-5p lost the ability to form spheroids and loosely accumulated (Figure 4E). When treated with Vemurafenib the spheroid size of A375 scr control cells and A375 miR-129-5p knockdown cells shrank and the surface became smoother. Spheroid area was reanalyzed 72 h, 96 h and 144 h after Vemurafenib treatment. A375 cells with miR-129-5p knockdown showed a significant stronger and faster spheroid growth compared to A375 control cells (Figure 4E; Appendix A).

Our results reveal miR-129-5p as a tumor suppressor in melanoma and that its repression attenuates BRAFi response.

### 3.5. SOX4 is a Targeted by miR-129-5p During BRAFi Response

To understand the molecular function of miR-129-5p, we performed in silico analyses to predict potential targets mRNAs of genes, which were known to be involved in melanoma progression and drug resistance. One potential target of miR-129-5p is *SOX4*, which was reported to induce cell proliferation [47] and to mediate BRAF inhibitor resistance in melanoma [48]. *SOX4* harbors three putative miR-129-5p binding sites in its mRNA 3′ UTR (Figure 5A). When we analyzed *SOX4* expression after BRAF inhibitor treatment, we found an inverse correlation of *SOX4* mRNA and SOX4 protein levels. *SOX4* mRNA was significantly increased (Figure 5B), while the protein levels were decreased after BRAF inhibition in A375 and WM35 cell lines (Figure 5C). To verify that SOX4 protein decrease caused by Vemurafenib was mediated by induction of miR-129-5p, we transfected A375 cells with a miRZip-129 construct, to block miR-129-5p function. Treatment of those cells with Vemurafenib still decreased SOX4 protein, however blocking miR-129-5p through miRZip-129 partially attenuated SOX4 reduction caused by Vemurafenib (Figure 5D). Since EZH2 inhibition induced miR-129-5p expression, we hypothesize that EZH2 inhibition mediates SOX4 protein level. In support of this hypothesis we found that treatment of A375 and WM35 cells with EZH2 inhibitor resulted in decreased SOX4 protein levels (Figure 5E). To identify the function of miR-129-5p in post-transcriptional *SOX4* regulation, we overexpressed miR-129-5p in A375 and WM35 cells. In Western blot analyses, we determined a decreased SOX4 protein level in miR-129-5p overexpressing melanoma cell lines (Figure 5F).

To prove the direct binding of miR-129-5p to the *SOX4* 3′UTR we performed luciferase reporter assays in A375 cells. Luciferase activity decreased after miR-129-5p overexpression (Figure 5G), which indicates the direct binding of miR-129-5p to the *SOX4* 3′UTR.

To validate that SOX4 influences emergence of BRAFi resistance, we measured SOX4 protein levels in BRAFi sensitive (A375, WM35) and resistant (A375R, WM35R) cells. In western blot analyses BRAFi resistant cells showed an increased SOX4 protein expression compared to the parental cells (Figure 5H). Furthermore, SOX4 knockdown by siRNA resulted in an improved BRAF inhibitor response of BRAFi resistant cells (A375R sictrl IC_50_: 33.8 µM vs. A375R siSOX4 IC_50_: 23.2 µM) (Figure 5I).

Thus, these results demonstrate that SOX4 is a direct target of miR-129-5p during melanoma response to BRAFi and they indicate that SOX4 mediates resistance to BRAFi treatment.

## 4. Discussion

We showed by both in silico and in vitro analyses that miR-129-5p is induced by BRAFi or MEKi treatment exclusively in melanoma cell lines with BRAF mutations, and not in primary normal human epidermal melanocytes (NHEMs), BRAF wildtype melanoma cells or BRAFi resistant melanoma cells. This BRAFi-mediated increase of miR-129-5p expression is then dynamically reduced with the emergence of resistance. Mechanistically, we were able to show that the miR-129-5p repression was mediated by EZH2, a downstream effector of BRAF. Our study provides evidences that miR-129-5p acts as a tumor suppressor improving the response to BRAFi and inhibiting the proliferation of melanoma cells by targeting *SOX4*. Due to these results we conclude that miR-129-5p is an important molecular regulator of response to BRAFi, which is repressed by downstream signaling pathways of constitutively active BRAF in melanoma.

The expression of several miRNAs in melanoma is altered by BRAFi and MEKi treatment [41,49,50] and reprogramming of miRNA expression is involved in the emergence of drug resistance in BRAF mutated melanoma [51]. Thus far, the miR-204, miR-211, as well as miR-410-3p, were shown to be induced by BRAF or MEK inhibitor and to contribute to resistance to BRAF inhibitor in melanoma by enhancing the activity of downstream pathways, such as MEK or ERK [39,40,50]. In contrast to these three miRNAs, miR-129-5p, was not only induced during response to BRAFi or MEKi treatment, but also decreased when the cells developed resistance. In addition without treatment miR-129-5p expression was reduced in BRAFi resistant cell lines compared to parental cells. These results were also supported by our in silico findings, that miR-129 has comparable expression levels in samples of progressive melanomas both prior to treatment with Dabrafenib or Vemurafenib and at time of tumor progression (GSE50509 dataset). Our results thus reveal that miR-129-5p expression correlates strongly with the response to BRAFi/MEKi in melanoma.

Previous studies showed that miR-129 is downregulated in melanoma tissues [52] and several different cancer entities compared to normal tissues [43,53,54,55,56,57]. In breast cancer [43], endometrial cancer [42] and gastric cancer [44] repression was mediated by epigenetically modifications of H3K27 trimethylation or DNA methylation. Both processes are regulated by the epigenetic modifier EZH2 [45,46]. EZH2 expression is associated with high proliferation rates and aggressive tumor subgroups of cutaneous melanoma [58]. It controls melanoma growth and metastasis through silencing of distinct tumor suppressors [59,60]. Of note EZH2 was also shown as a mediator of treatment resistance in BRAF mutated melanoma [61,62]. We have now demonstrated that, in melanoma, EZH2 represses miR-129-5p, dependent on constitutive active BRAF signaling. After treatment of BRAF mutated melanoma cells with BRAFi or MEKi, we observed a reduction of EZH2 protein as well as mRNA, as shown previously for melanoma [62,63]. Additionally, specific inhibition of EZH2 induces miR-129-5p expression in BRAF mutated melanoma cell lines, independent from their BRAFi response status, while BRAF wildtype cells and normal melanocytes show no changes of expression. In silico, miR-129 and EZH2 expression during treatment correlated inversely with administration of BRAF inhibitor (GSE98314). This correlation could not be determined in the melanoma cohort of the cancer genome atlas (TCGA, https://www.cancer.gov/tcga, accessed on 22 April 2021), because these patients had not received BRAFi or MEKi. However, miR-129-5p expression was elevated in BRAF mutated melanoma patients harboring EZH2 missense or silencing mutations compared to wildtype EZH2. This provides further evidence that, in melanoma, EZH2 mediates miR-129-5p expression downstream of constitutive active BRAF signaling. Although previous studies showed a reduction of H3K27me3 during treatment with BRAFi or MEKi in melanoma [62,63] we could not confirm these changes in our experiments. These different results may be explained by using low doses of treatment (0.4 µM Vemurafenib or 10 nM Trametinib) or shorter treatment times (48 h). Treatment with the cytosine analogue Cytarabine (AraC), which inhibits DNA methylation, a process also mediated by EZH2 [45,59], induces miR-129-5p expression. Our results and the previously described repression of miR-129 via EZH2 in different entities by DNA methylation or histone modification [43,64] let us conclude, that in BRAF mutated melanoma suppression of miR-129-5p is mediated via EZH2 downstream of constitutive active BRAF signaling.

Our functional analyses demonstrate that miR-129-5p inhibits cell proliferation, cell cycle progression and that it mediates BRAFi response in BRAF mutated melanoma. This is in line with previous in vitro studies reporting that miR-129-5p acts as tumor suppressor in lung cancer [65], gastrointestinal cancer [55,57,66,67], esophageal squamous cell carcinoma [68,69], hepatocellular cancer [70], cervical cancer [71], breast cancer [43,72], and glioblastoma [73].

We reveal that miR-129-5p overexpression improves response to BRAFi of resistant melanoma cells, while knockdown of miR-129-5p in parental cells shows the opposite effect. These results are supported by our finding that miR-129-5p expression is decreased during emergence of resistance. While other studies also found that repression of miR-129-5p modulates resistance to multiple drugs (e.g., chemotherapeutic, such as 5-Fluorouracil, Gemcitabine and Adriamycine, as well as antibody therapy, such as Trastuzumab) in several cancers, e.g., gastric cancer [44], breast cancer [43,74,75], ovarian cancer [76], and bladder cancer [77], our study, for the first time, reports on miR-129-5p as mediator of BRAFi response in melanoma.

Having elaborated that miR-129-5p functions as tumor suppressor and mediates BRAF inhibitor treatment response, we investigated potential target genes, known to be involved in melanoma progression and drug resistance. *SOX4* seemed a likely fitting candidate, since it is not only induced during melanoma progression and promotes melanoma proliferation by AKT signaling activation [47], but also mediates to BRAFi in melanoma through regulation of IGF-1R [48]. Our results reveal that BRAFi treatment induces *SOX4* mRNA, but reduces SOX4 protein, which implicates a posttranscriptional regulation. By a luciferase reporter assay, we confirmed *SOX4* as direct target of miR-129-5p. *SOX4* was also observed to be a target of miR-129-5p in breast cancer [43], esophageal carcinoma [69], chondrosarcoma [54] or cervical cancer [71], in which miR-129-5p mediated repression of SOX4 was associated with reduced cancer cell progression.

We then demonstrated that knockdown of *SOX4* by applying a siRNA resulted in an improved response of BRAFi resistant cells towards Vemurafenib treatment, comparable to miR-129-5p overexpression. This result indicates that miRNA-mediated mechanisms of resistance could indeed be amenable to therapeutic modifications, e.g., by agomiRs or mimics. However, tissue distribution and targeted cell delivery is still an obstacle for the systemic therapeutic approach [31,32].

Additional putative targets of miR-129-5p that are associated with drug resistance in cancers are ABC transporters (ABCB1, ABCC5, ABCG1) [44] and RUNX1 [78]. For RUNX1 we could not prove a posttranscriptional regulation by miR-129-5p in BRAF associated melanoma (data not shown). Further research should investigate if ABC transporters act as additional targets of miR-129-5p in the context of BRAF/MEK inhibitor resistance in melanoma.

Since SOX4 controls EZH2 expression by direct promotor binding [79] and SOX4/EZH2 are shown to interact as co-repressors on tumor suppressive miR-31 in invasive esophageal cancer cells [80], we hypothesize a regulatory network in BRAF mutated melanoma, in which constitutively activated BRAF signaling induces SOX4 and EZH2 expression, resulting in miR-129-5p repression (Figure 6). During BRAF/MEK inhibition, EZH2 is reduced, which releases the epigenetic repression of miR-129-5p. The elevated miR-129-5p inhibits SOX4 protein translation, resulting in a reduction of proliferation and improved treatment response (Figure 6). The emergence of resistance by reactivation of BRAF downstream or bypass pathways reinforces EZH2 resulting in repression of miR-129-5p.

Taken together our results emphasize SOX4 as a potential therapeutic target in BRAF driven melanoma which could be attacked by pharmaceutically, e.g., by miR-129-5p mimics.

## 5. Conclusions

Targeted therapies, e.g., BRAFi improved patient outcomes in BRAF mutated melanoma. Emergence of resistance to these therapies is an obstacle urgent to overcome. To this end, we wanted to investigate whether miRNAs enhance response to BRAF inhibition. In this study we identified the tumor suppressor miR-129-5p to be induced during BRAF inhibition. Finally, we found that miR-129-5p decreases melanoma cell proliferation and improves response to BRAFi by targeting *SOX4*.

Taken together our results emphasize miR-129-5p, as well as SOX4, as potential therapeutic targets in BRAF-driven melanoma.

## Figures and Tables

**Figure 1 cancers-13-02393-f001:**
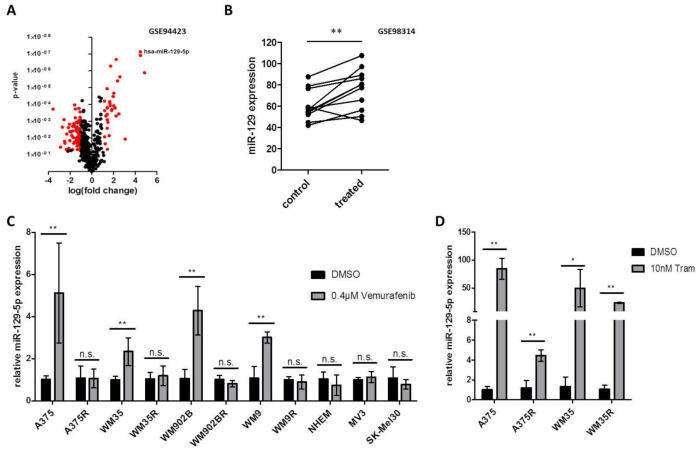
miR-129-5p expression is increased during BRAFi and MEKi treatment. (**A**) Volcano plot of miRNAs differentially expressed in BRAFi treated A375 cells vs. a group of three different conditions: A375 treated with DMSO and a resistant clone of A375 cells (A375R) treated with Vemurafenib or DMSO (GSE94423). Red dots represent significant differentially expressed miRNAs (log_2_ fold change: ≥1; ≤−1; *p* ≤ 0.05). (**B**) Analyses of the miR-129 expression in eleven BRAF mutated melanoma cell lines treated with DMSO (control), Dabrafenib or a combination of Dabrafenib and Trametinib (treated). Expression data are shown as cubic spline function normalized values. (**C**) qRT-PCR analyses of miR-129-5p after BRAFi (0.4 µM Vemurafenib, 24 h) in BRAF mutation associated cell lines (A375, WM35, WM902B, WM9), BRAFi resistant cell lines (A375R, WM35R, WM902BR, WM9R), normal human epidermal melanocytes (NHEM) and BRAF wildtype melanoma cell lines (MV3, SK-MEL30). (**D**) qRT-PCR for miR-129-5p expression after MEK inhibition by 10 nM Trametinib for 48 h on parental sensitive (A375, WM35) and the corresponding resistant BRAF mutated cell lines (A375R, WM35R). Bars represent average ± standard deviation of at least three independent experiments. * *p* ≤ 0.05; ** *p* ≤ 0.01.

**Figure 2 cancers-13-02393-f002:**
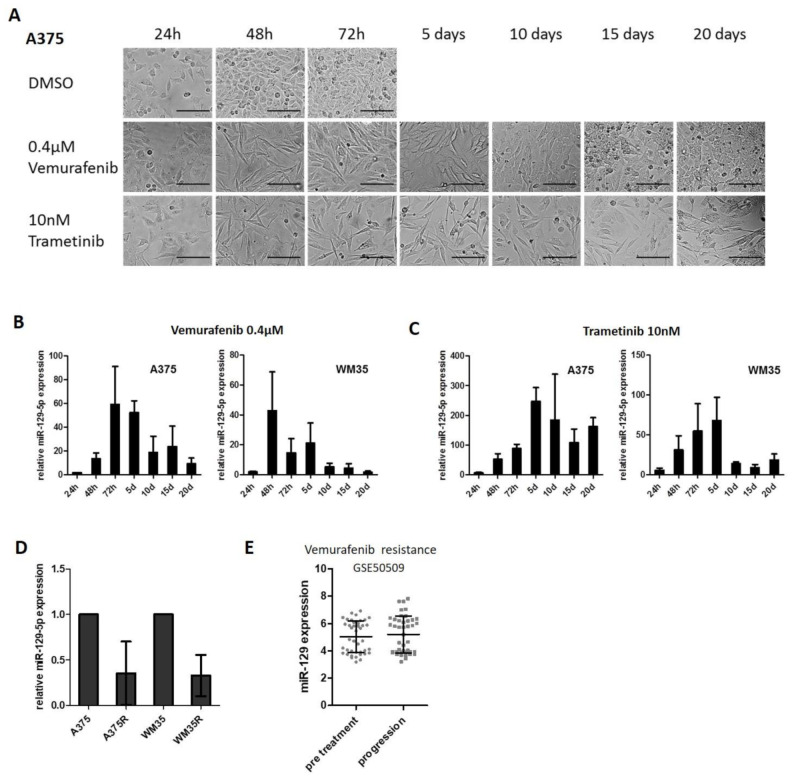
miR-129-5p expression decreases during emergence of resistance to BRAFi. (**A**) A375 cells were treated with Vemurafenib or Trametinib every second day for a period of 20 days. Images were taken at indicated time points. Scale bars represent 200 µm. (**B**,**C**) Expression of miR-129-5p was analysed by qRT-PCR during permanent BRAF (**B**) or MEK (**C**) inhibition over 20 days in two BRAF mutated melanoma cell lines (A375 and WM35). (**D**) qRT-PCR expression analyses of miR-129-5p in A375/WM35 compared to the corresponding BRAFi resistant cell line (A375R/WM35R). (**E**) Analysis of miR-129-5p expression in the data set GSE50509 comparing untreated melanoma samples and samples at tumor progression during BRAFi treatment. Bars represent average ± standard deviation of at least three independent experiments.

**Figure 3 cancers-13-02393-f003:**
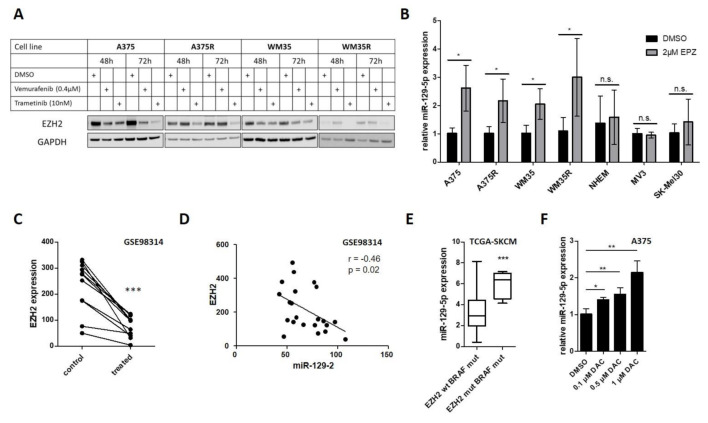
miR-129-5p is repressed by BRAF/EZH2 pathway. (**A**) Western blot analyses of EZH2 in sensitive and resistant A375 and WM35 cells after BRAF and MEK inhibition. GAPDH immunoblot was used for normalization. (**B**) miR-129-5p expression of BRAF mutated cell lines (A375, WM35), corresponding BRAFi resistant cell lines (A375R, WM35R), normal human epidermal melanocytes (NHEM) and BRAF wildtype melanoma cell lines (MV3, SK-Mel30) after inhibition of EZH2 by EPZ (48 h). (**C**) Analysis of EZH2 expression in 11 BRAF mutated melanoma cell lines after BRAFi or BRAFi/MEKi compared to DMSO treated cells (control) (GSE98314). (**D**) Pearson correlation of EZH2 and miR-129-5p expression (GSE98314). Data are presented as cubic spline function normalized values. (**E**) Comparison of miR-129-5p expression in BRAF mutated melanoma patients with and without EZH2 mutation (TCGA). Values are presented as log2(RPM+1). (**F**) qRT-PCR analyses of miR-129-5p expression in A375 cells after DNA methylation inhibition by decitabine for 72 h. Bars represent average ± standard deviation of at least three independent experiments. n.s., not significant; * *p* ≤ 0.05; ** *p* ≤ 0.01; *** *p* ≤ 0.001.

**Figure 4 cancers-13-02393-f004:**
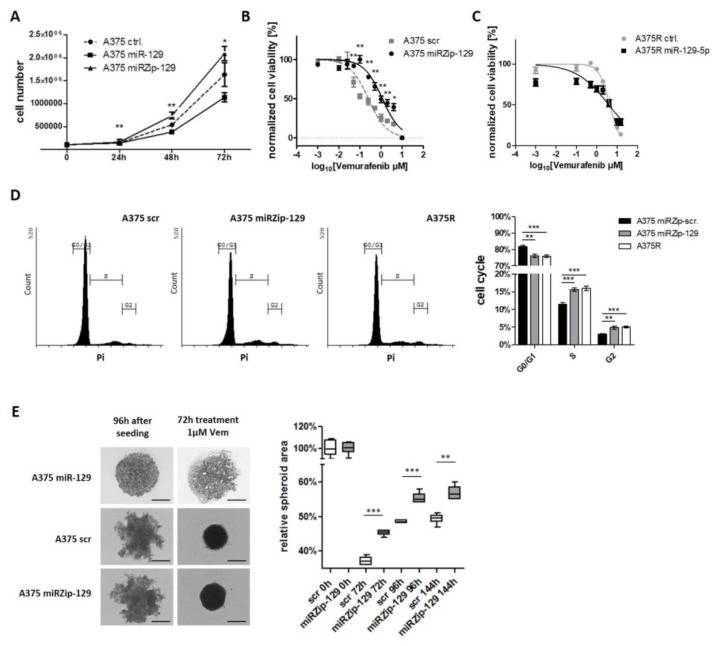
miR-129-5p acts as a tumor suppressor. (**A**) Proliferation Assay of A375 cells overexpressing miR-129-5p (A375 miR-129) and knockdown miR-129-5p (A375 miRZip-129) compared to parental A375 (control). (**B**) BRAFi treatment response of miR-129-5p knockdown cells (A375 miRZip-129) and (**C**) BRAFi resistant cells overexpressing miR-129-5p (A375R miR-129) using cell viability assay. (**D**) Cell cycle distribution of sensitive parental A375 cells and their corresponding Vemurafenib resistant clone (A375R) compared to miR-129-5p knockdown cells (A375 miRZip-129). (**E**) Representative photographs of spheroid growth after initial Vemurafenib treatment (1 µM) were taken at indicated time points. Scale bar represents 500 µm. Spheroid area was measured of at least 5 spheroids using ImageJ software. Bars represent average of at least three independent experiments. * *p* ≤ 0.05; ** *p* ≤ 0.01; *** *p* ≤ 0.001.

**Figure 5 cancers-13-02393-f005:**
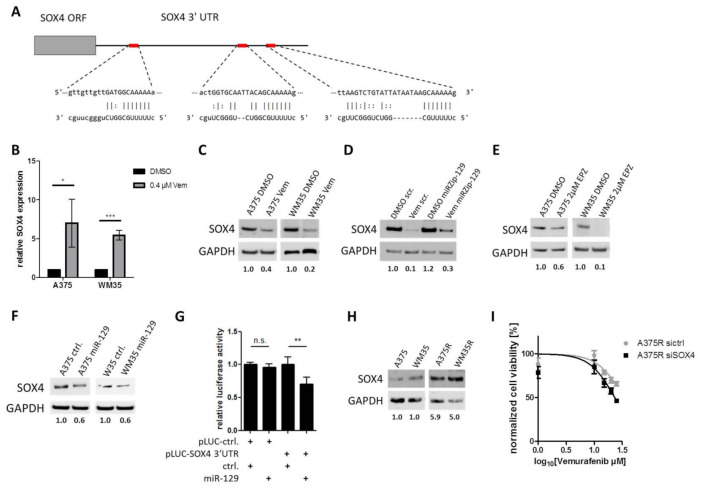
miR-129-5p targets *SOX4*. (**A**) miR-129-5p and its three predicted binding sites at the 3’UTR in *SOX4*. (**B**) *SOX4* mRNA expression and (**C**) protein expression after BRAFi treatment in A375 and WM35 cells. (**D**) SOX4 protein level was compared in BRAFi treated (Vem) or untreated (DMSO) A375 cells with miR-129-5p knockdown (miRZip-129) or scramble control (scr). (**E**) Western blot of SOX4 protein in A375 or WM35 cells after EZH2 inhibition by EPZ (48 h; 2 µM). (**F**) SOX4 protein levels in A375 and WM35 cells overexpressing miR-129-5p or scramble control. (**G**) Luciferase reporter assay was performed in A375 cells. Cells were co-transfected with *SOX4* 3′UTR luciferase reporter construct (pLuc-SOX4 3′UTR) or control (pLuc-ctrl.) and miR-129-5p expression construct (miR-129) or control (ctrl.) respectively. Luciferase activity was analysed after 24 h. (**H**) Western blot analyses of SOX4 in parental and corresponding BRAFi resistant A375 and WM35 cells. (**I**) Viability assay of BRAFi treated A375R cells after *SOX4* knockdown by siRNA or control. Cells were treated for 96 h. Bars represent average of at least three independent experiments. * *p* ≤ 0.05; ** *p* ≤ 0.01; *** *p* ≤ 0.001.

**Figure 6 cancers-13-02393-f006:**
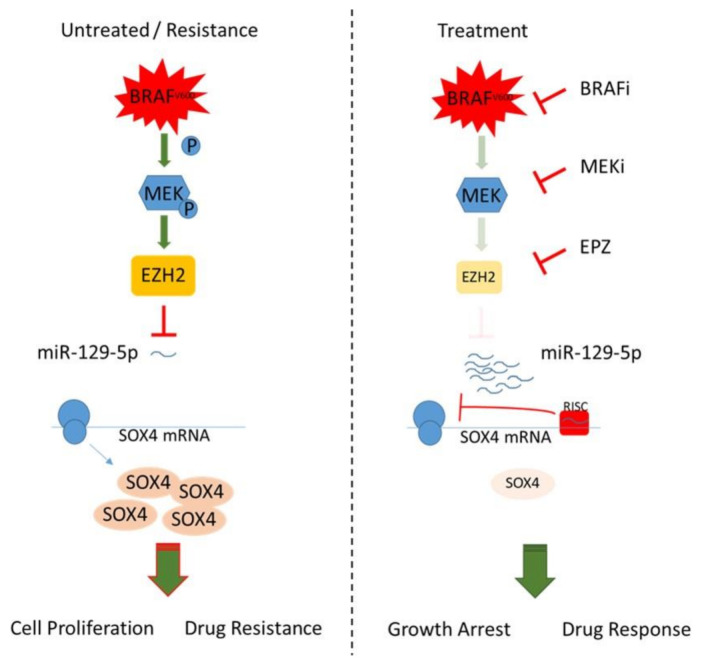
BRAF/EZH2 signaling represses miR-129-5p inhibition of SOX4 thereby modulating BRAFi resistance in melanoma.

## Data Availability

The data presented in this study are available in this article and Appendix A.

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
