# Peer review of "BRAF/EZH2 Signaling Represses miR-129-5p Inhibition of SOX4 Thereby Modulating BRAFi Resistance in Melanoma"

_cancers, 2021, doi:10.3390/cancers13102393_

Round 1
Reviewer 1 Report
The authors show that treating BRAF mutant melanoma with Braf or mek inhibitors induce the expression of miR-129-5p. This induction is absent in cell lines resistant to Braf inhibitors when treated with inhibitors of Braf but not inhibitors of MEK. The increase in miR-129-5p in sensitive cell lines correlates with a decrease in proliferation and a decrease in EZH2 expression. Inhibiting EZH2 increases miR-129-5p, and increased miR-129-5p blocks upregulation of SOX4 following Braf inhibitor treatment. There does not appear to be a correlation between levels of miR-129-5p and disease progression, suggesting that the upregulation of miR-129-5p is an early event following Brafi treatment.
Why do the authors conclude that BRAF/MEK signaling drives miR-129-5p expression? miR-129-5p is increased when BRAF/MEK signaling is inhibited. In the discussion section, the authors say that inhibition of BRAF/MEK signaling induces miR-129-5pm expression.
Why did the authors choose to treat cells with either a Braf or Mek inhibitor and not use the combination therapy that is currently used in the clinic? What happens to the expression of miR-129-5pm when cells are treated with the combination of Braf and Mek inhibitors?
Does the upregulation of miR-129-5p correlate with cell death following Braf or Meki or is it simply a growth arrest? If targeting SOX4 induces a growth arrest and not cell death with BRAF or MEK inhibitors, how would targeting SOX4 create a sustainable response in patients?
Minor comments
Fig3B. What time point was used for treatment with EZH2 inhibitor?
Figure 4B. Are these curves statistically different?
Fig. 4C. How are is the A375R miR-129-5p cell line normalized at 0 uM? The whole data set looks like it’s shifted down compared to the A375 control.
Fig 4I. Why do the IC50 curves stop at 50% or less viability? Why didn’t the authors use higher concentrations of Vemurafenib to generate a full curve?
Reviewer 2 Report
The manuscript shows interesting results regarding EZH2/miR-129/SOX regulatory axis in melanoma and its involvement in response to BRAF inhibition in melanoma and drug resistance, however, it suffers several important issues that need to be addressed before considering to be published in Cancers journal. All these issues will be pointed out below:
- the involvement of miR-204 and miR-211 in resistance to BRAF inhibition in melanoma is highly controversial, as questioned by Vitiello et al. (Oncoscience, 2018). The action of both miRNAs in melanoma is highly context-dependent and numerous reports state their tumor suppressive roles in melanoma. Therefore the Introduction and Discussion fragments should be rewritten;
- what did the authors mean by "manufacture's introduction" on line 182?
- line 216: "Having demonstrated that miR-129-5p expression is mediated by BRAF/MEK signaling," this statement is not true, the authors show only the positive correlation between BRAFmut and expression of miR-129-5p in BRAFmut melanomas. To prove that miR-129-5p is indeed mediated by BRAF/MEK, additional siRNA experiments should be performed;
- line 217-218: what did the authors mean by "We treated A375 cells or WM35 cells with 217 Vemurafenib or Trametinib every second day for a period of 20 days"?
- cytarabine is primarily used as a non-specific inhibitor of DNA synthesis and hence S-phase arrest inducer. The use of cytarabine as an inhibitor of DNA methylation is inappropriate. Why did not the authors use potent and specific DNA methylation inhibitors, such as azacitidine, decitabine or zebularine?
- which antibody was used for H3K27me3 Western Blots? This information is missing in Methods section.
- did the authors perform EZH2 siRNA experiments to definitely confirm that the expression of miR-129 is indeed EZH2-dependent?
- line 313-314: "Luciferase activity decreased after miR-129-5p overexpression (Figure 5G), which indicates the 313 direct binding of miR-129-5p to the SOX4 3’UTR." This is also only a hypothesis. To address this issue, more complex Luciferase experiments should have been planned (ie. checking either and both miRNA-binding sequences in SOX4 3'-UTR, with appropriate mutant miR-129 mimetics)
- most of the experiments are accompanied with very high sd error bars, that severely impact the importance and veracity of described results. This is clearly visible on Figure 2 b, c and d, Figure b and f and Figure 5 b (A375 bar). The authors should reexamine these experiments with more repeats, so that these results are more accurate
- Figure 2 e should be reorganized, so that median and error bars are clearly visible;
- Figure 3 c, d and e: what are the units on y-axis?
- Figure 4 d (bars) I basically don't see any differences in cell cycle in all three situations that would be important to draw conclusions about any cell cycle arrest.
- an additional Figure 6, that schematically shows the described regulatory axis would be very handy as a graphical summary of the whole paper
- the manuscript needs moderate English improvement: several typos and grammar mistakes needs to be addressed.
Round 2
Reviewer 2 Report
The response to reviewer's remarks has been prepared in details and almost all issues denoted by the reviewer have been addressed.